# Progressive Lower Extremity Paralysis Caused by Intrathecal MTX-Induced Myelopathy Mimicking Guillain–Barre Syndrome: A Case Report

**DOI:** 10.3390/diagnostics13213337

**Published:** 2023-10-30

**Authors:** Eun Chae Lee, Dong Gyu Lee

**Affiliations:** Department of Physical Medicine and Rehabilitation, College of Medicine, Yeungnam University, Daegu 42415, Republic of Korea; dldmsco13@naver.com

**Keywords:** intrathecal, methotrexate, myelopathy, Guillain–Barre syndrome

## Abstract

Methotrexate (MTX) is commonly used in intrathecal chemotherapy for patients with acute lymphocytic leukemia (ALL) to prevent central nervous system (CNS) involvement. However, the use of MTX-based chemotherapy can lead to rare yet severe complications, such as MTX-induced myelopathy. Here, we report the case of MTX-induced myelopathy initially misdiagnosed as Guillain–Barre syndrome, leading to a delay in diagnosis and treatment. We present a case of a 39-year-old male with a history of B-cell acute lymphoblastic leukemia (B-ALL) who experienced bilateral foot paresthesia and progressive lower extremity weakness after intrathecal methotrexate (MTX) treatment. Initially, the patient was suspected as having Guillain–Barre syndrome (GBS) due to similar clinical features and nerve conduction studies. The patient received intravenous immunoglobulin (IVIG) treatment, but his condition worsened. T2-weighted images of the thoracic spinal cord revealed high signal intensity in both lateral and posterior columns, typically associated with subacute combined degeneration. However, elevated vitamin B12 levels ruled out SCD in this case. Based on the aforementioned findings, intrathecal methotrexate-induced myelopathy was diagnosed. This case highlights the diagnostic challenge posed by the similarity in clinical presentation between MTX-induced myelopathy and GBS. Differentiating between these conditions is critical for appropriate management. Prompt recognition and treatment with folate metabolism compounds may mitigate neurological sequelae.

In acute lymphocytic leukemia (ALL) patients, intrathecal (IT) chemotherapy is known to reduce the risk of central nervous system (CNS) leukemia [1]. However, rare but serious complications can arise after methotrexate (MTX)-based chemotherapy, such as methotrexate-induced myelopathy, which can lead to significant long-term sequelae [2,3]. 

Guillain–Barre syndrome (GBS) is an acute polyneuropathy that typically progresses from the lower limbs to the upper limbs following an infection [4]. Nerve conduction studies have played a crucial role in diagnosing and distinguishing the subtypes of GBS, aiding in long-term prognosis prediction [5]. Although most cases start in the lower limbs and progress to the upper limbs, some cases may present initially as lower limb weakness, potentially leading to initial misdiagnosis as myelopathy [6]. The association between GBS and ALL has not been definitively established. In patients with ALL, GBS can arise due to immunosuppression following intensive chemotherapy [7], and immune system neoplasms can induce AIDP (acute inflammatory demyelinating polyneuropathy). AIDP is characterized by inflammation and damage to the protective myelin sheath of peripheral nerves, leading to weakness, numbness, and paralysis [8]. In some rare cases, GBS can occur after a stem cell transplantation due to various factors [9].

Here, we report the case of MTX-induced myelopathy initially misdiagnosed as GBS.

A 39-year-old male patient, diagnosed with chronic myeloid leukemia (CML) in June 2010 and later with B-cell acute lymphoblastic leukemia (B-ALL) in August 2017, presented with bilateral sole paresthesia in December 2022. He received intrathecal MTX and blinatumomab treatment for six months from September 2021 to March 2022 due to relapsed B-ALL. For 6 months, he was administered intrathecal chemotherapy with 15 mg methotrexate 18 times. Because of testis invasion of B-ALL, he received involved site radiation therapy (ISRT) on testis. However, B-ALL relapsed again after six months, resulting in the administration of inotuzumab, intrathecal MTX, and cytarabine for three months from September 2022 to December 2022. For 3 months, he was administered intrathecal chemotherapy with 15 mg methotrexate 14 times and 50 mg cytarabine 11 times. 

In December 2022, the patient visited our clinic with bilateral foot paresthesia and gradually progressive weakness in the lower extremities that persisted for a week following intrathecal MTX treatment. Neurological examination revealed manual muscle testing (MMT) grade 4 in both lower limbs. A nerve conduction study (NCS) conducted at the time revealed prolonged F-wave latency in bilateral peroneal nerves and the left tibial nerve, slightly slow motor nerve conduction velocity in the right sural and superficial peroneal nerves, low compound motor action potential (CMAP) in bilateral peroneal nerves, and low sensory compound nerve action potential (CNAP) in bilateral peroneal nerves. These findings indicate a diffuse sensory–motor peripheral neuropathy with a predominant axonopathy. A follow-up evaluation of muscle strength conducted seven days later showed an MMT grade of 2- at bilateral lower limb. Additionally, nerve conduction study results no significant changes compared to the previous study. Considering the clinical presentation and electromyographic findings, acute motor and sensory axonal neuropathy (AMSAN) variant of GBS was suspected.

Consequently, intravenous immunoglobulin (IVIG) treatment was administered for five days, but muscle strength evaluation eight days later indicated no improvement. Eventually, the muscle strength in both lower limbs deteriorated to MMT zero, with no sensation of muscle contraction. Additionally, the patient complained of neuropathic pain, decreased temperature sensation, reduced vibratory sensation, and impaired proprioception below the T6 level, so spine MRI was taken. T2-weighted images of the thoracic spinal cord revealed high signal intensity in both lateral and posterior columns (Figure 1). 

In a laboratory test, vitamin B12 levels were elevated (1183), as were folate levels (>40 ng/mL). The MRI findings of subacute combined degeneration (SCD) typically display a distinct symmetrically increased signal intensity in the posterior column on T2-weighted images, as seen in this patient. This condition arises from vitamin B12 deficiency; however, in this case, SCD could be excluded, as the patient had elevated vitamin B12 levels. Unfortunately, a cerebrospinal fluid (CSF) study could not be performed at the time. Considering all of the above findings, a diagnosis of intrathecal methotrexate-induced myelopathy was established. Therefore, the intravenous immunoglobulin (IVIG) treatment was stopped. Intravenous vitamin B12 supplementation was administered and rehabilitation was continued. Five months later, there was no improvement in symptoms and motor recovery. The NCS follow-up examination revealed the absence of CMAP responses for both the peroneal and tibial nerves, as well as the absence of SNAP responses for both superficial peroneal nerves. The remaining conduction velocities were within normal limits. Ultimately, the patient progressed to paraplegia.

The exact mechanism by which MTX induces neurotoxicity remains unclear, but several hypotheses exist. First, MTX functions as an antimetabolite by inhibiting dihydrofolate reductase (DHFR), preventing the conversion of folate to tetrahydrofolic acid (THF) and causing intracellular folate deficiency [10]. Additionally, vitamin B12 and 5-methyltetrahydrofolate are necessary for the conversion of homocysteine to S-adenosylmethionine (SAM), but DHFR inhibition leads to SAM deficiency [10,11]. SAM plays a crucial role in the formation and maintenance of myelin sheaths; its deficiency can lead to demyelination.

MRI findings in patients with MTX-induced myelopathy typically exhibit high signal intensity on T2-weighted images in the posterior column, closely resembling the characteristics of subacute combined degeneration [12]. Similarly, our case displayed predominantly high signal intensity on T2-weighted images in the posterior and lateral columns, consistent with previous reports. Subacute combined degeneration (SCD) is a progressive spinal cord disorder characterized by lower limb weakness, sensory disturbances, and motor deficits due to vitamin B12 deficiency. Symmetrically increased signal intensity in the posterior column on T2-weighted MRI images is a hallmark of SCD. A condition resembling subacute combined degeneration was also taken into consideration, considering the findings from spinal MRIs. However, this patient exhibited elevated levels of vitamin B12, and the administration of vitamin B12 supplementation did not result in significant improvement. Therefore, the diagnosis of IT methotrexate-induced myelopathy seemed to be a more appropriate diagnosis than subacute combined degeneration.

Recent reports have presented the rapid recovery of lower limb paralysis in MTX-induced myelopathy patients with the administration of various folate metabolism compounds, such as SAM, folic acid, and cyanocobalamin [11,13]. In patients with CTX-induced myelopathy, medications such as steroids or IVIG, in conjunction with vitamin B12, can be used [14]. Moreover, the administration of dextromethorphan to patients with severe methotrexate-induced CNS toxicity has demonstrated symptomatic improvement [15,16]. Methotrexate (MTX) disrupts the remethylation process of homocysteine into methionine, resulting in elevated homocysteine levels. Homocysteine metabolites act as NMDA agonists, contributing to neurotoxicity. Consequently, the noncompetitive NMDA receptor agonist, dextromethorphan, holds promise in attenuating toxic NMDA receptor stimulation in MTX-induced neurotoxicity. While cyanocobalamin administration did not lead to improvement in our case, considering other cases, prompt diagnosis and aggressive folate therapy might have potentially mitigated neurological sequelae. Ultimately, when patients who have undergone intrathecal MTX chemotherapy present with complaints of progressive paralysis, it is important to promptly differentiate spinal cord pathology.

## Figures and Tables

**Figure 1 diagnostics-13-03337-f001:**
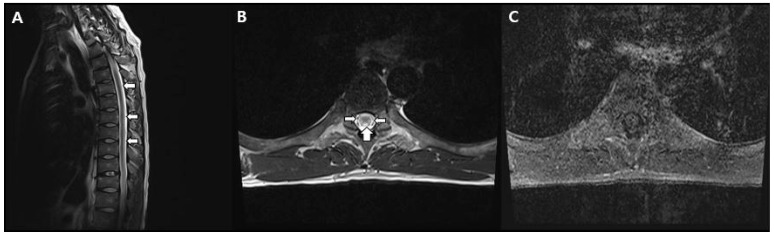
Spine magnetic resonance imaging findings. (**A**) The T2-weighted sagittal image shows high signal in the thoracic cord, long segment (arrow). (**B**) The T2-weighted axial image shows signal hyperintensity in bilateral lateral and posterior columns, mainly white matter (arrow). (**C**) The T1-weighted axial image with contrast enhancement did not reveal any enhanced lesions within the spinal cord.

## Data Availability

Data are available from the authors upon reasonable request.

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
