# Peer review of "Progressive Lower Extremity Paralysis Caused by Intrathecal MTX-Induced Myelopathy Mimicking Guillain–Barre Syndrome: A Case Report"

_diagnostics, 2023, doi:10.3390/diagnostics13213337_

Round 1

Reviewer 1 Report

Comments and Suggestions for Authors

This is an interesting case report on Progressive Lower extremity Paralysis caused by intrathecal methotrexate-induced myelopathy mimicking Guillain-Barre syndrome. It will add information to the literature about this complication. I would recommend this manuscript to be accepted for publication after minor revision.

There are some suggestions to the authors.

1. Authors can describe how many doses of intrathecal methotrexate and cytarabine were given to the patient in the treatment regimen.

2. Authors may discuss any other treatment option other than folate replacement. Some case reports showed clinical improvement to steroid therapy.

3. Some updated references can be added. For example, page3, line 109 “Recent reports suggest rapid recovery of lower limb paralysis in MTX-induced myelopathy patients with the administration of various folate metabolism compounds, such as SAM, folic acid, and cyanocobalamin”. Those recent reports can be quoted in the references.

4. Page 2 line 64, “the patient complained of decreased pain” instead of “the patient complained decreased pain

Comments on the Quality of English Language

The overall quality of English is satisfactory.

Author Response

We are submitting revision of our manuscript (diagnostics-2669620), of which the title is “Progressive Lower Extremity Paralysis Caused by Intrathecal MTX-Induced Myelopathy mimicking Guillain-Barre Syndrome: A Case Report” for your kind consideration of its suitability for publication in the Diagnostics.

We appreciate the insightful and helpful comments of the editor and reviewers very much. We have made as many changes as possible according to the editor’s and reviewer’s recommendations and have prepared the responses in a point-by-point fashion. We hope that our revision is satisfactory to the standards of the editors and reviewers and look forward to hearing the ultimate decision.

#1 reviewer

Authors can describe how many doses of intrathecal methotrexate and cytarabine were given to the patient in the treatment regimen.

Author’s response

Thank you for your specific and valuable suggestion. I fully agree with your comment. As you pointed out, we have included the dosage and frequency of intrathecal methotrexate and cytarabine administered to the patient in lines 59-60 and 54-55 Thank you.

Authors may discuss any other treatment option other than folate replacement. Some case reports showed clinical improvement to steroid therapy.

Author’s response

I appreciate your kind and specific comment. As you mentioned As you mentioned, additional information regarding using other medications has been provided in lines 120-127.

Some updated references can be added. For example, page3, line 109 “Recent reports suggest rapid recovery of lower limb paralysis in MTX-induced myelopathy patients with the administration of various folate metabolism compounds, such as SAM, folic acid, and cyanocobalamin”. Those recent reports can be quoted in the references.

Author’s response

I am always grateful for your comments. I have searched for other recent reports and added them as Reference 11.13. Thank you.

Page 2 line 64, “the patient complained of decreased pain” instead of “the patient complained decreased pain”

Author’s response

I greatly appreciate your kind comment. I have made the adjustments to the English expression as you suggested. Thank you.

Reviewer 2 Report

Comments and Suggestions for Authors

The authors compiled a short case presentation with interesting images. In my opinion, it should be considered for publication after the addressing the following issues:

-  “A subsequent spine MRI revealed MTX-induced myelopathy” is misleading. Actually, the MRI revealed signal changes suggestive for myelopathy and that’s all. These are later to be clinically defined/integrated in the diagnosis of  MTX-induced myelopathy. Please rephrase.

- Please define AIDP

- “…. lymphoblastic leukemia (B-ALL) in August 2017, presented with bilateral sole paresthesia” When?

- Please state the MTX dose per administration and exact number of MTX administrations.

- “…patient complained decreased pain” ? Please rephrase for better clarity.

- T2 images are nice but you should consider adding DWI and post contrast sequences, especially because CSF analysis is unavailable.

- Additionally, patient medical history should clearly rule out any other cause of myelopathy such as radiation therapy .

Comments on the Quality of English Language

Minor English editing is required to some highlighted sentences.

Author Response

We are submitting a revision of our manuscript (diagnostics-2669620), of which the title is “Progressive Lower Extremity Paralysis Caused by Intrathecal MTX-Induced Myelopathy mimicking Guillain-Barre Syndrome: A Case Report” for your kind consideration of its suitability for publication in the Diagnostics.

We appreciate the insightful and helpful comments of the editor and reviewers very much. We have made as many changes as possible according to the editor’s and reviewer’s recommendations and have prepared the responses in a point-by-point fashion. We hope that our revision is satisfactory to the standards of the editors and reviewers and look forward to hearing the ultimate decision.

#2 reviewer

“A subsequent spine MRI revealed MTX-induced myelopathy” is misleading. Actually, the MRI revealed signal changes suggestive for myelopathy and that’s all. These are later to be clinically defined/integrated in the diagnosis of MTX-induced myelopathy. Please rephrase.

Author’s response

Thank you for your delicate comment. As you pointed out, It seems that the original sentence could be misleading. I have revised and supplemented the content in lines 19-23. Thank you.

Please define AIDP

Author’s response

Thank you for your comment. Adding a definition of AIDP would likely be more helpful for readers. I have included the aforementioned information in lines 40-43. Thank you.

“…. lymphoblastic leukemia (B-ALL) in August 2017, presented with bilateral sole paresthesia” When?

Author’s response

Thank you for your specific comment. The information you pointed out is highly crucial. I have added this content in line 48. Thank you.

Please state the MTX dose per administration and exact number of MTX administrations.

Author’s response

Thank you for your specific and good direction suggestion. As you mentioned, the dosage and frequency of intrathecal methotrexate and cytarabine administered to the patient have been added in lines 59-60, 63-64. Thank you.

“…patient complained decreased pain” ? Please rephrase for better clarity.

Author’s response

I greatly appreciate your kind comment. I have made the adjustments to the English expression as you suggested. Thank you.

T2 images are nice but you should consider adding DWI and post contrast sequences, especially because CSF analysis is unavailable.

Author’s response

Thank you for your comment. It would have been valuable if DWI was available, but unfortunately, DWI was not obtained. There were no enhancing lesions observed in the post-contrast sequence. I have added T1-weighted axial image with contrast enhancement on figure.

Additionally, patient medical history should clearly rule out any other cause of myelopathy such as radiation therapy.

Author’s response

Thank you for your comment. The information you highlighted is indeed crucial. It's important to note that our patient received Involved-Site Radiation Therapy (ISRT) to the testis, not the spinal cord. I have included this clarification in lines 51-52. Thank you.

Round 2

Reviewer 2 Report

Comments and Suggestions for Authors

Nice work. Just a minor issue:

- please define radiotherapy in common medical terminology, such as " 30 Gy total dose (2 Gy/fraction, 15 fractions)" or just delete the dose notice and leave ISRT to the testis.  

Author Response

I have made the corrections as you suggested, which include deleting the dose notice and leaving ISRT to the testis.

Thank you!